# Variability between Different Hand-Held Dynamometers for Measuring Muscle Strength

**DOI:** 10.3390/s24061861

**Published:** 2024-03-14

**Authors:** William Du, Kayla M. D. Cornett, Gabrielle A. Donlevy, Joshua Burns, Marnee J. McKay

**Affiliations:** 1Sydney School of Health Sciences, Faculty of Medicine and Health, The University of Sydney, Sydney, NSW 2006, Australia; widu3619@alumni.sydney.edu.au (W.D.); kayla.cornett@sydney.edu.au (K.M.D.C.); gabrielle.donlevy@sydney.edu.au (G.A.D.); marnee.mckay@sydney.edu.au (M.J.M.); 2Paediatric Gait Analysis Service of New South Wales, Sydney Children’s Hospitals Network, Westmead, NSW 2145, Australia

**Keywords:** load cell dynamometer, isometric muscle strength, outcome measures, clinical outcome assessments, clinical trials and cohort studies

## Abstract

Muscle strength is routinely measured in patients with neuromuscular disorders by hand-held dynamometry incorporating a wireless load cell to evaluate disease severity and therapeutic efficacy, with magnitude of effect often based on normative reference values. While several hand-held dynamometers exist, their interchangeability is unknown which limits the utility of normative data. We investigated the variability between six commercially available dynamometers for measuring the isometric muscle strength of four muscle groups in thirty healthy individuals. Following electro-mechanical sensor calibration against knowns loads, Citec, Nicholas, MicroFET2, and Commander dynamometers were used to assess the strength of ankle dorsiflexors, hip internal rotators, and shoulder external rotators. Citec, Jamar Plus, and Baseline Hydraulic dynamometers were used to capture hand grip strength. Variability between dynamometers was represented as percent differences and statistical significance was calculated with one-way repeated measures ANOVA. Percent differences between dynamometers ranged from 0.2% to 16%. No significant differences were recorded between the Citec, Nicholas, and MicroFET2 dynamometers (*p* > 0.05). Citec grip strength measures differed to the Jamar Plus and Baseline Hydraulic dynamometers (*p* < 0.01). However, when controlling for grip circumference, they were comparable (*p* > 0.05). Several hand-held dynamometers can be used interchangeably to measure upper and lower limb strength, thereby maximising the use of normative reference values.

## 1. Introduction

Muscle strength is routinely assessed in patients with neuromuscular diseases such as Charcot–Marie–Tooth neuropathy, Duchenne muscular dystrophy, and spinal muscular atrophy to identify the musculoskeletal consequences of muscle atrophy and to evaluate the impact of therapeutic interventions. Muscle weakness associated with these disorders can result in pain, disability, contractures, and falls. As such, accurate, reliable, and sensitive tools for evaluating muscle force are required to identify the severity of weakness, monitor progression, and evaluate treatment response [1].

Manual muscle testing using the Medical Research Council grading system has been used extensively in clinical practice, clinical trials, and cohort studies. While this method of muscle strength assessment has acceptable reliability when implemented by experienced clinical evaluators [2], the subjective categorisation of muscle strength demonstrates significant limitations in detecting small yet clinically significant changes [3,4]. To differentiate between varying degrees of muscle weakness more precisely, instrumented load cell methods such as fixed dynamometry (e.g., Cybex, CSMi, HUMAC NORM, Stoughton, MA, USA).) and hand-held dynamometry (e.g., Citec, CIT Technics, Groningen, The Netherlands) are often used [5]. Fixed dynamometry involves large force transducers which are considered the gold standard in measuring muscle strength. However, high cost (>USD 50,000) and lack of portability limits their clinical applicability [6]. In comparison, hand-held dynamometers are portable devices usually incorporating a wireless load cell with an inbuilt microprocessor to measure isometric muscle force with good to excellent reliability across both clinical and healthy populations [1,7,8]. These characteristics have made hand-held dynamometers a feasible and widely used tool to quantify muscle strength in a range of clinical populations such as those with inherited neuropathy [9,10], spinal muscular atrophy [11,12], osteoarthritis [13,14], sarcopenia [6], or inflammatory myopathies [15,16]. Increasingly, muscle strength measured by hand-held dynamometry is being used as primary and secondary endpoints in clinical trials of disease-modifying therapies in neuromuscular and musculoskeletal disorders.

To quantify disease severity and evaluate the meaningful therapeutic effect using hand-held dynamometry, a comparison with age- and sex-matched normative reference values is commonly undertaken. Normative values are especially important for patients with bilateral strength impairments, where changes or differences in muscle strength cannot be easily identified by using the non-affected limb as a reference. Numerous normative datasets exist; however, there is considerable variability in sample size (n = 31 to 1000), muscle groups (upper and/or lower limb), model of hand-held dynamometer, and methodology variations such as the size and circumference of the hand grip device [17]. Normative reference values exist for several hand-held dynamometers including MicroFET2 (Hoggan Scientific, LLC, Salt Lake City, UT, USA) [18,19], Nicholas (Lafayette Instrument Company, Lafayette IN, USA) [20], Commander (JTECH Medical, Salt Lake City, UT, USA) [21], Accuforce II (Ametek, Largo, FL, USA) [22], and Citec (CIT Technics, Groningen, The Netherlands) [23,24]. However, the varied characteristics of these hand-held dynamometers (e.g., unit of measurement, upper force limit, device design) have restricted the comparison of these different normative reference datasets and limited their utility because it is not known if measures of strength assessed using different hand-held dynamometers are comparable [4]. Despite the availability of many different hand-held dynamometers, few studies have evaluated the variability of these different devices to infer interchangeability. Much of the literature evaluating the variability of dynamometers has focused on hand grip strength [25,26], assessed a single muscle group [27], compared only two models [28], or used device fixation which is not always clinically feasible [27].

Different models of hand-held dynamometers are used to assess muscle strength in clinics around the world. Device selection can be influenced by cost, geographic supplier restrictions, and personal preference. For multi-site international clinical trials, a particular model is usually mandated to ensure comparability of data between sites. This results in extensive start-up costs to purchase new hand-held dynamometer models for consistency instead of using the model available at each site. Establishing whether muscle strength values recorded on one hand-held dynamometer are interchangeable with measures recorded on a different hand-held dynamometer will improve the utility of the existing normative reference values and assist clinicians to diagnose pathology and gauge the effectiveness of therapeutic interventions targeting muscle weakness.

Therefore, the aim of this study was to evaluate the variability between six commercially available hand-held dynamometers for measuring the isometric muscle strength of four major muscle groups in thirty healthy individuals.

## 2. Materials and Methods

### 2.1. Study Design and Participants

The study design was a single-group repeated measures cross-sectional study where each participant was tested with each of the six hand-held dynamometers in accordance with the standardised 1000 Norms Project Protocol for isometric muscle strength testing [29]. Recruitment was carried out within the Greater Sydney metropolitan area in Australia, using structured convenience sampling techniques, including advertising via e-newsletters, community flyers, and through word of mouth. Eligible participants were aged 18 years or older, considered themselves healthy for their age, and could participate in age-appropriate activities of daily living. People with significant health conditions affecting physical performance, for example neuropathic, inflammatory, or degenerative conditions, as well as those with infectious or inflammatory arthropathies were excluded.

Thirty healthy adults volunteered to participate in this study, 50% were female, and they were aged 18 to 57 years (mean 29.2, SD 10.9 years). A sample size of n = 30 was deemed sufficient to assess the reliability of hand-held dynamometry in accordance with previous studies using similar methods [28,30,31,32]. The study received institutional ethics approval from the Human Research and Ethics Committee (HREC 2018/181) and informed written consent was obtained from all participants.

### 2.2. Equipment

The isometric muscle strength of ankle dorsiflexors, hip internal rotators, shoulder external rotators, and hand grip were measured using six hand-held dynamometers. Muscle groups were selected to widely represent the proximal and distal muscle groups of the upper and lower limb. The hand-held dynamometers used to measure shoulder, hip, and ankle strength were the Citec (CIT Technics, Groningen, The Netherlands), Nicholas (Lafayette Instrument, Lafayette, IN, USA), MicroFET2 (Hoggan Scientific, LLC, Salt Lake City, UT, USA), and the Commander (JTECH Medical Industries, Inc., Midvale, UT, USA). The hand-held dynamometers used to assess grip strength were the Citec, Jamar Plus (Performance Health Supply, Cedarburg, WI, USA), and Baseline Hydraulic (Fabrications Enterprises, White Plains, NY, USA). The manufacturer details and measurement specifications of each hand-held dynamometer are specified in Table 1.

### 2.3. Calibration

Each hand-held dynamometer was calibrated to determine if they accurately measured static forces. The bespoke calibration set up was designed to specifically accommodate the electro-mechanical sensor of each hand-held dynamometer, or the hydraulic system for the Baseline Hydraulic hand-held dynamometer (Figure 1). A wooden beam was secured by clamps to a metal frame one metre off the ground. Each hand-held dynamometer was placed in the middle of the wooden beam and stabilised by a metal jig, specifically designed to accommodate each different model. A length of aluminium channel was centred on top of the force transducer of each hand-held dynamometer. A strap ran along the channel and encircled both the hand-held dynamometer and the beam, allowing known loads to be suspended from it. Each known load was placed on top of a trolley of adjustable height. The trolley was raised, moved under the hand-held dynamometer, and slowly lowered to apply the load. Each hand-held dynamometer was calibrated with increasing known loads applied to the centre of their load cell. Loads ranging from 30 to 415 N, selected based on the ranges of normative muscle strength values quantified in the 1000 Norms Project [23], were progressively added to the load cell of each hand-held dynamometer and the displayed reading was recorded. For each known load, the measurement was recorded twice, and the mean value determined. Measurements of at least four known loads were recorded for each hand-held dynamometer across its measurement range. Known loads were quantified using the Mettler ID1 Multi Range Industrial Balance 120 kg (August Sauter, GmbH Albstadt 1 for Mettler Instrumente; calibrated as per the manufacturer’s instructions).

### 2.4. Data Collection

A standardised protocol for positioning, collection, and recording was adopted for each of the six hand-held dynamometers. Each participant was assessed using the ‘make’ technique to measure muscle strength, whereby the clinical evaluator acts as a fixed point to meet the participant’s maximal force. Muscle strength tests were performed in gravity neutralised positions for all muscle groups on their dominant limb. Following a practice, each participant was instructed to perform three maximal voluntary contractions of ankle dorsiflexors, hip internal rotators, shoulder external rotators, and hand grip lasting 3 to 5 s each. A rest period of 15 s between each contraction and device was given. The Citec hand-held dynamometer has one grip setting, whilst the Baseline Hydraulic and Jamar Plus have adjustable grip positions. When hand grip strength using Baseline Hydraulic and Jamar Plus was collected, the adjustable grip was set so that fingertips reached the proximal palmar crease, as per each device manufacturer’s instructions. In accordance with the Citec manufacturer’s instructions, the displayed grip values were multiplied by two, as the grip applicator measures strength in a 1:2 ratio.

The order of the hand-held dynamometers was randomised to minimise the influence of a learning effect or fatigue. The Excel RAND (Excel, Microsoft Corporation, Redmond, WA, USA) function was used to generate a randomised hand-held dynamometer testing order. The four muscle groups were also assessed in a random order with one hand-held dynamometer before testing with the other devices. Standardised verbal encouragement was used and the measures for each muscle group were recorded. The clinical evaluator was blinded to the muscle strength values by another investigator who documented and obscured the hand-held dynamometer recording using a cardboard cover during testing. The average of three trials using each device was calculated. In a pilot study to test the standardised protocol in 10 participants aged 20–53 years (mean 33.1, SD 10.5 years), the intrarater reliability of the clinical evaluator (W.D.) was established as acceptable for all four muscle groups (ICC_3,1_ 0.78–0.95).

### 2.5. Isometric Muscle Strength Assessment Protocol

To assess ankle dorsiflexion, participants were positioned lengthways sitting with their feet off the edge of an examination table. The clinical evaluator stabilised the participant’s lower leg against the examination table with one hand and positioned the ankle in mid-range and then placed the hand-held dynamometer against the dorsal surface of the foot just proximal to the metatarsal heads. The clinical evaluator instructed the participant to “push against the device as if trying to bend your foot up towards you as hard as you can for several seconds. Ready, set, go”. See Figure 2.

To assess hip internal rotation, participants were positioned on an examination table upright sitting with their hips and knees in 90° of flexion with their legs over the edge of the examination table, holding on to the edge of the examination table with both hands. The clinical evaluator placed the hand-held dynamometer on the lateral surface of the leg just proximal to the lateral malleolus and instructed the participant to “push against the device as if trying to swing your leg outwards as hard as you can for several seconds”. See Figure 3.

To assess shoulder external rotation, participants were seated upright in a chair, feet supported, shoulders in neutral, elbows in 90° flexion, and forearms in neutral. The clinical evaluator placed the hand-held dynamometer against the extensor surface of the forearm, just proximal to the wrist crease and instructed the participant to “push against the device as if trying to rotate your arm outwards as hard as you can for several seconds”. See Figure 4.

To assess hand grip, participants were seated comfortably in a chair with their feet supported. The hand and forearm being assessed was positioned with the shoulder adducted and in neutral rotation, with the elbow in 90° of flexion, forearm in neutral, and the wrist in 0 to 30° of extension and 0 to 15° of ulnar deviation. The participant was asked to grasp the hand-held dynamometer with the fingers wrapped around the handle. The clinical evaluator instructed the participant to perform a maximal contraction lasting three to five seconds, saying, “when I say go, I want you to squeeze the handle as hard as you can”. See Figure 5.

### 2.6. Data Analysis

Statistical analysis was performed in SPPS, Version 28 (IBM SPSS Statistics for Win-dows, Armonk, NY, USA). For calibration, a scatterplot was constructed for each hand-held dynamometer against known loads to determine measurement accuracy. A line (y = x) was constructed to indicate the ideal fit. Variability between hand-held dynamometers was represented as percent differences and their statistical significance was calculated with one-way repeated measures analysis of variance (ANOVA) with pairwise comparisons. A significant difference was considered if *p* < 0.05.

## 3. Results

### 3.1. Calibration

The scatterplot of force measures recorded by the six hand-held dynamometers against known loads is shown in Figure 6. Each calibration point’s proximity to the y = x ideal curve demonstrated measurement accuracy for all devices in measuring known forces.

### 3.2. Variability

Muscle strength values collected using the six hand-held dynamometers were obtained from thirty participants (50% female) aged 18 to 57 years (mean 29.2, SD 10.9 years). Percent differences between hand-held dynamometers for measuring ankle dorsiflexors, hip internal rotators, and shoulder external rotators ranged from 0.2% to 5.9% and there were no significant differences between the muscle strength measurements assessed using the Citec, Nicholas, and MicroFET2 hand-held dynamometers (*p* > 0.05). The Commander under-recorded across all muscle groups; however, it was only significant for hip internal rotation (Table 2). Pairwise comparisons identified that the Commander demonstrated a significant difference to the Citec (*p* = 0.049) and Nicholas (*p* = 0.021) when measuring hip internal rotation, with percent differences calculated as 5.9% (Citec) and 5.3% (Nicholas).

Percent differences between the Citec, Jamar Plus, and Baseline Hydraulic hand-held dynamometers for measuring hand grip strength ranged from 1.3% to 16.0% (Table 2). There was no significant difference between the grip strength measurements assessed using the Baseline Hydraulic and Jamar Plus hand-held dynamometers; however, the Citec grip measures were significantly lower than Jamar Plus and Baseline Hydraulic by approximately 50 N (*p* < 0.01).

Post hoc evaluation of the different grip size settings (lever arm length) was conducted. While the circumference of the Citec grip applicator was constant at 13.2 cm, the Baseline Hydraulic and Jamar Plus hand-held dynamometers had five grip size options, with grip circumference ranging from 11.5 cm to 21.1 cm. A one-way repeated measures ANOVA was performed on a subgroup of eight participants (aged 20 to 37 years; seven females) who used the two smallest settings on the Baseline Hydraulic and Jamar Plus and Citec hand-held dynamometers (11.5 cm or 14 cm). Percent differences between the Citec, Jamar Plus, and Baseline Hydraulic hand-held dynamometers for measuring hand grip strength in this subgroup ranged from 2.1% to 4.7% (Table 2). There was no significant difference between any devices for hand grip strength when the smallest settings were used (*p* > 0.05).

## 4. Discussion

The Citec, Nicholas, and MicroFET2 hand-held dynamometers consistently measured the isometric muscle strength of the ankle dorsiflexors, hip internal rotators, and shoulder external rotators. For grip strength, the Citec, Baseline Hydraulic, and Jamar Plus were only consistent if the two smallest grip circumference settings of the Baseline Hydraulic and Jamar Plus were used. The Commander under-recorded muscle strength compared with the Citec, Nicholas, and MicroFET2. Given that the devices were calibrated against known loads, and the clinical evaluator was trained and reliable, it is unlikely that any measurement deviations could be due to technical or behavioural errors. Instead, these deviations are likely attributable to the inherent characteristics of each device.

Our finding that several different hand-held dynamometers can be used interchangeably to measure upper and lower limb muscle strength concurs with a previous study that investigated the Nicholas and MicroFET2 in 30 participants to measure hip, knee, and ankle strength, showing good to excellent inter-device agreement [32]. Prior research has highlighted inconsistencies among other hand-held dynamometers, including Kimura et al. [28] who concluded that the Chatillon CSD500 (Chatillon Medical Products, Greensboro, NC, USA) and MicroFET2 hand-held dynamometers were not interchangeable. However, the scope of their study was limited as they only evaluated knee extensors in 12 healthy young adults using the ‘break’ test, which requires the clinical evaluator to apply sufficient resistance to just overcome the force of the participant [33]. The ‘break’ test method is known to be less reliable and often underestimates muscle strength, due to the clinical evaluator’s inability to counteract knee extensor force and sustain the test position. Similarly, Fenter et al. [27] compared measures of hip abduction muscle strength between the MicroFET2, Jamar Hydraulic, and Dial Push-Pull Gauge in 10 healthy females aged 24–42 years and concluded that the MicroFET2 was poorly correlated with the Jamar Hydraulic and the Dial Push-Pull Gauge (Chatillon Medical Products, Greensboro, NC, USA). However, they acknowledged that since the MicroFET2 was fixed to a stabilising bar, it may have influenced the agreement between devices.

Despite equivalent calibration outcomes with known loads, there were significant differences between the Baseline Hydraulic, Jamar Plus, and Citec hand-held dynamometers for hand grip strength. Inter-device comparisons for grip strength in the literature are inconsistent. For example, the Jamar Hydraulic has demonstrated good inter-device consistency with the Roylan Hydraulic (Patterson Medical Supply Inc., Bolingbrook, IL, USA), BTE–Primus (BTE, Hanover, MD, USA), and MicroFET4 (Hoggan Scientific, LLC, Salt Lake City, UT, USA) [30,34,35]. However, when compared to the Jamar Plus and the Takei^®^ (TKK Model 5101 Digital, Takei Scientific Instruments, Niigata, Japan), the Jamar Hydraulic has been found to consistently over-record [25] and conversely under-record hand grip strength [36]. These conflicting results could be due to the Jamar Hydraulic, Jamar Plus, and Takei^®^ having between two and five possible grip circumference settings, and the specific grip setting used was not consistently reported. The Baseline Hydraulic hand-held dynamometer has previously demonstrated acceptable inter-device agreement with the Jamar Hydraulic hand-held dynamometer when set at the second smallest of five grip positions [25]. When we compared the subset of our participants who used the smallest grip setting on the Baseline Hydraulic, Jamar Plus, and Citec hand-held dynamometers, muscle strength measures were comparable, providing further evidence that lever arm length influences hand grip strength production [37]. In the future, reporting hand grip size in all studies is strongly encouraged.

This study revealed that the majority of the tested hand-held dynamometers can be used interchangeably for measuring isometric muscle strength, thereby enhancing their usefulness in both clinical and research settings. By benchmarking against available normative reference data of isometric muscle strength, clinicians and researchers can evaluate intervention efficacy and track disease progression, without needing the same hand-held dynamometer model. By enhancing the accessibility and applicability of normative muscle strength reference databases, the need to create separate normative reference databases for multiple devices is greatly reduced.

An example of how to improve the accessibility and applicability of muscles strength normative reference values is the online platform, www.ClinicalOutcomeMeasures.org, accessed on 16 January 2024. This platform provides free access to normative isometric muscle strength reference data collected from people aged between 3 to 100 years as part of the 1000 Norms Project. This web-based scoring system is used for monitoring responses to therapy and houses key clinical trial endpoints such as the Charcot-Marie-Tooth Pediatric Scale (CMTPedS), CMT Functional Outcome Measure (CMT-FOM), CMT Infant Scale (CMTInfS), and Rasch-modified CMT Neuropathy Score (CMTNSv2-R). With the understanding that different hand-held dynamometers can be used interchangeably, clinicians and researchers worldwide can input their data into these calculators to assess therapeutic efficacy without the need to use the same hand-held dynamometer model. Understanding hand-held dynamometer interchangeability broadens the use for these measurement devices, making them more accessible and applicable globally.

This study is not without limitation. First, we only tested four muscle groups. However, we ensured that one proximal and one distal muscle group of the upper and lower limb was selected as sufficient proof of concept. Further, the chosen muscle groups also spanned a variety of strength magnitudes, ranging from ~90 N to ~250 N. Second, although we tested six widely used hand-held dynamometers, we did not test all commercially available devices and the results may not be generalisable beyond the tested devices. Third, only eight individuals used the smallest setting of the Baseline Hydraulic and Jamar Plus hand-held dynamometers resulting in a small sample size for this subgroup analysis.

## 5. Conclusions

The Citec, Nicholas, and MicroFET2 hand-held dynamometers were comparable for measuring ankle dorsiflexors, hip internal rotators, and shoulder external rotators. For hand grip strength, the Citec, Jamar Plus, and Baseline Hydraulic hand-held dynamometers were comparable if the smallest grip setting was used. This study suggests that several hand-held dynamometers can be used interchangeably to measure upper and lower limb muscle strength, thereby maximising the use of existing normative reference values.

## Figures and Tables

**Figure 1 sensors-24-01861-f001:**
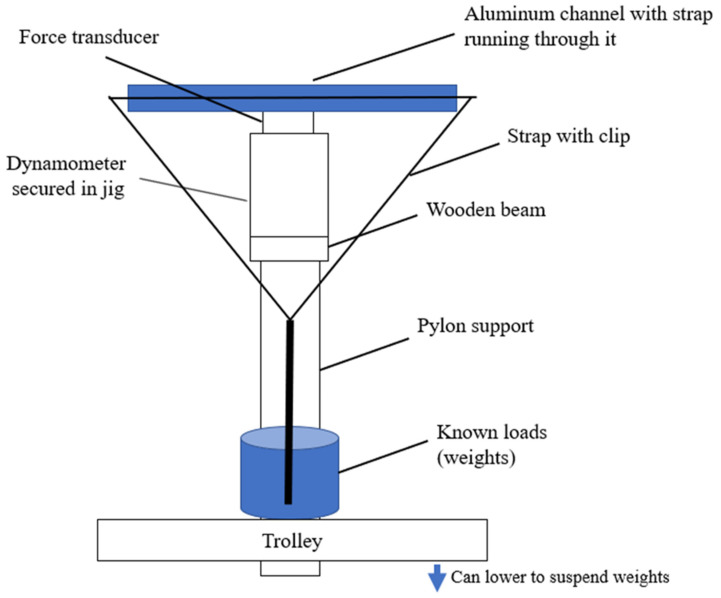
Hand-held dynamometer calibration set up, illustrative image without scaling.

**Figure 2 sensors-24-01861-f002:**
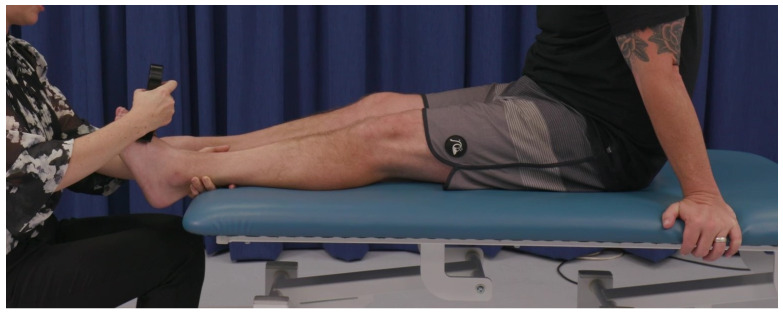
Assessment of ankle dorsiflexion isometric muscle strength.

**Figure 3 sensors-24-01861-f003:**
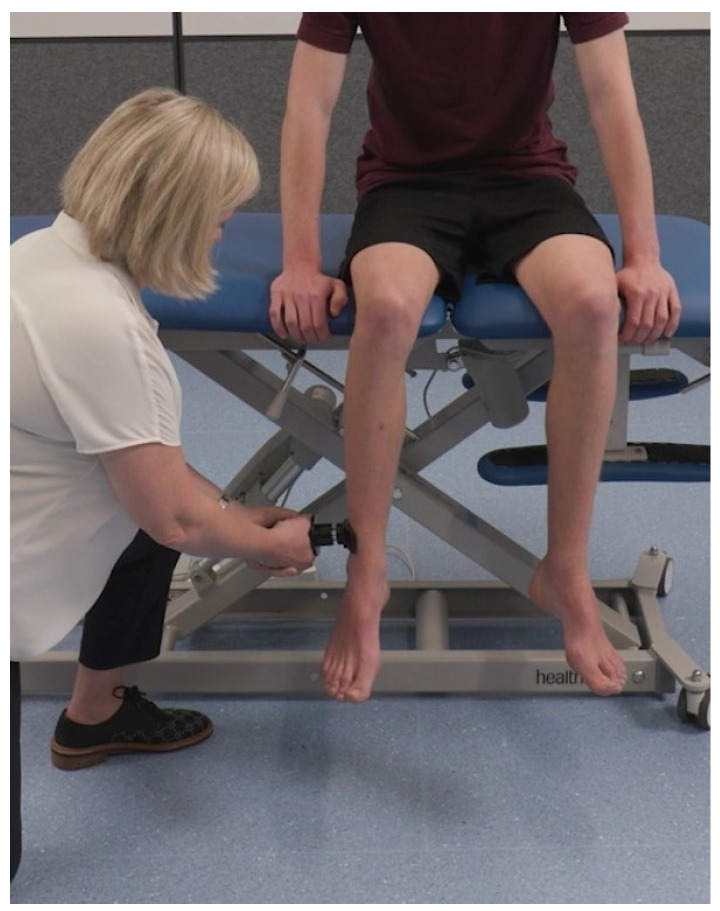
Assessment of hip internal rotation isometric muscle strength.

**Figure 4 sensors-24-01861-f004:**
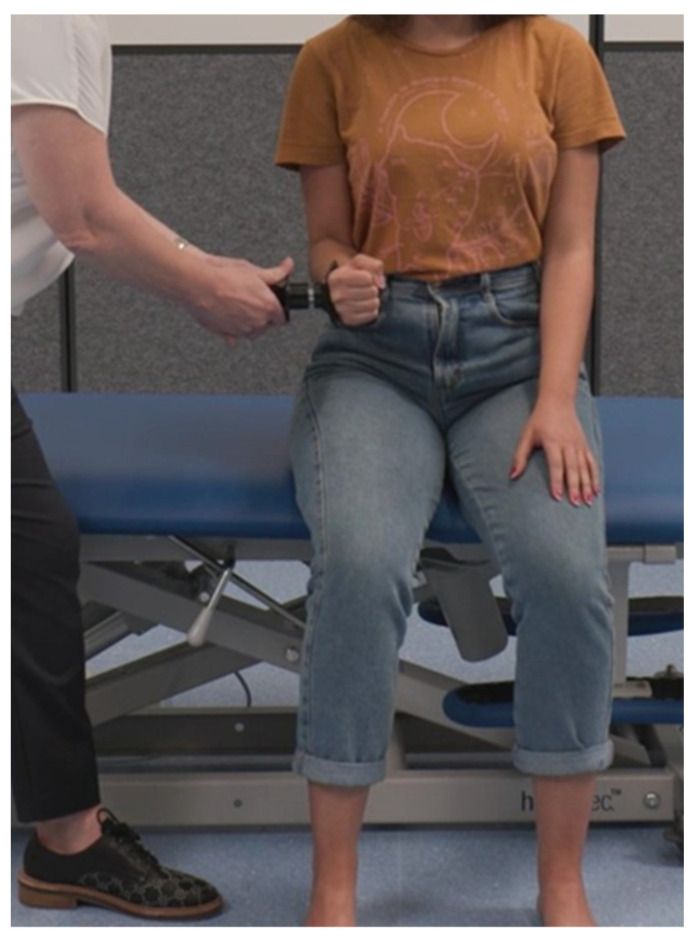
Assessment of shoulder external rotation isometric muscle strength.

**Figure 5 sensors-24-01861-f005:**
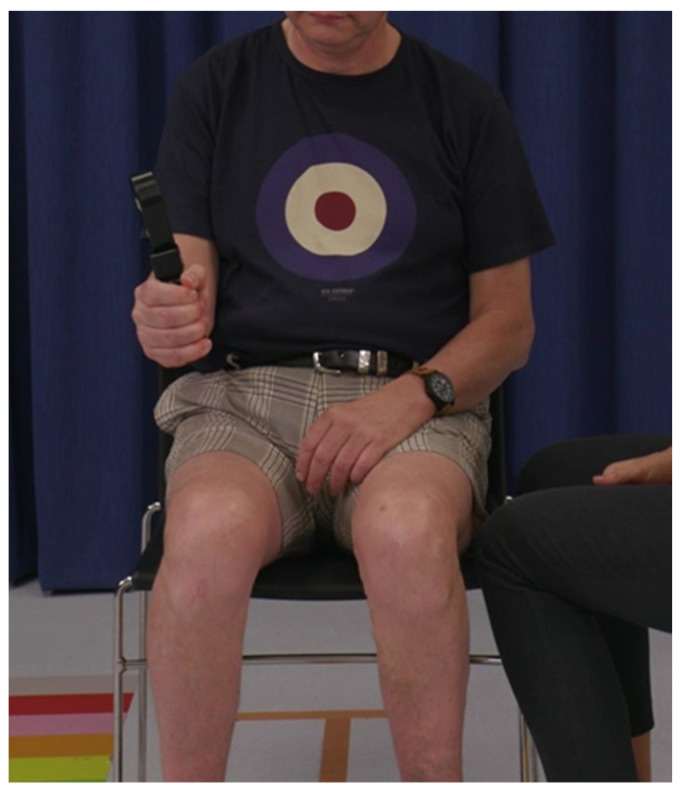
Assessment of hand grip muscle strength.

**Figure 6 sensors-24-01861-f006:**
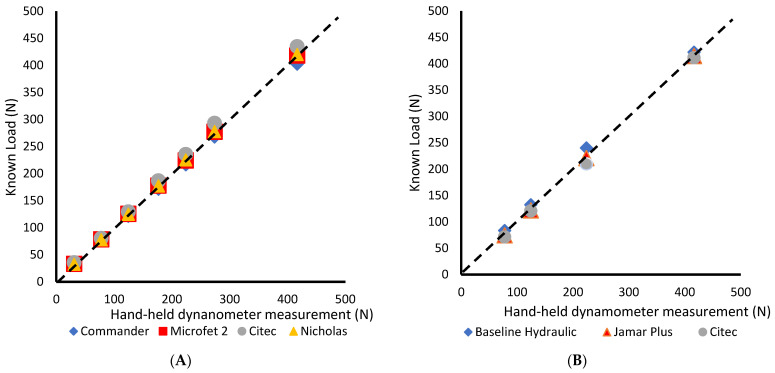
Comparison of the six hand-held dynamometers against known loads, (**A**) Hand-held dynamometers used to measure ankle, hip, and shoulder muscle strength; (**B**) Hand-held dynamometers used to measure hand grip strength.

**Table 1 sensors-24-01861-t001:** Details of the six hand-held dynamometers used in this study.

Dynamometer	Manufacturer	Unit Range	Muscle Groups
Citec Hand-held Dynamometer CT 3001 (Citec)	CIT Technics, Groningen, The Netherlands	0–500 Newtons	Hand grip muscles Shoulder external rotators Hip internal rotators Ankle dorsiflexors
Lafayette Model 01163, Nicholas Manual Muscle Tester (Nicholas)	Lafayette Instrument, Lafayette, IN, USA	0–1335 Newtons	Shoulder external rotators Hip internal rotators Ankle dorsiflexors
Hoggan MicroFET2 (MicroFET2)	Hoggan Scientific, LLC, Salt Lake City, UT, USA	0–660 Newtons	Shoulder external rotators Hip internal rotators Ankle dorsiflexors
Commander PowerTrack II Muscle Tester (Commander)	JTECH Medical Industries, Inc. Midvale, UT, USA	0–550 Newtons	Shoulder external rotators Hip internal rotators Ankle dorsiflexors
Baseline Hydraulic Hand Dynamometer, Model 10602 (Baseline Hydraulic)	Fabrications Enterprises, White Plains, NY, USA	0–890 Newtons	Hand grip muscles
Jamar Plus Digital Hand Dynamometer (Jamar Plus)	Performance Health Supply, Cedarburg, WI, USA	0–890 Newtons	Hand grip muscles

**Table 2 sensors-24-01861-t002:** Variability between six different hand-held dynamometers for measuring upper and lower limb muscle strength (n = 30).

Muscle Group and Device	Mean (SD)	Range	Mean (SD) and Percentage Difference (%)				
			Citec	Nicholas	Commander		MicroFET2
**Shoulder External Rotation (N)**							
Citec	98.1 (30.1)	59.7–169.0		2.6 (10.6) 2.7%	4.9 (11.2) 5.1%		2.8 (8.1) 2.9%
Nicholas	95.5 (26.2)	63.1–169.7	2.6 (10.6) 2.7%		2.3 (7.8) 2.4%		0.2 (8.0) 0.2%
Commander	93.2 (26.9)	55.0–165.0	4.9 (11.2) 5.1%	2.3 (7.8) 2.4%			2.1 (7.8) 2.2%
MicroFET2	95.3 (27.7)	60.2–160.0	2.8 (8.1) 2.9%	0.2 (8.0) 0.2%	2.1 (7.8) 2.2%		
**Hip Internal Rotation (N)**							
Citec	168.9 (50.4)	90.0–276.0		0.9 (18.5) 0.6%	9.7 (18.7) 5.9% ^a^		2.2 (24.1) 1.3%
Nicholas	167.8 (42.6)	96.8–242.6	0.9 (18.5) 0.6%		8.7 (15.0) 5.3% ^a^		1.3 (21.2) 0.6%
Commander	159.2 (40.6)	94.1–242.0	9.7 (18.7) 5.9% ^a^	8.7 (15.0) 5.3% ^a^			7.5 (21.8) 4.5%
MicroFET2	166.7 (47.8)	97.8–264.5	2.2 (24.1) 1.3%	1.3 (21.2) 0.6%	7.5 (21.8) 4.5%		
**Ankle Dorsiflexion (N)**							
Citec	254.6 (53.5)	111.0–359.7		8.0 (32.8) 3.1%	3.4 (33.7) 1.3%		6.3 (25.1) 2.4%
Nicholas	262.6 (52.3)	122.3–350.1	8.0 (32.8) 3.1%		11.3 (29.3) 4.4%		1.7 (28.4) 0.6%
Commander	251.3 (44.5)	165.7–338.3	3.4 (33.7) 1.3%	11.3 (29.3) 4.4%			9.6 (25.2) 3.7%
MicroFET2	260.9 (52.2)	127.7–338.5	6.3 (25.1) 2.4%	1.7 (28.4) 0.6%	9.6 (25.2) 3.7%		
**Hand Grip (N)**			**Citec**	**Jamar Plus**		**Baseline Hydraulic**	
Citec	284.0 (83.9)	149.3–457.3		45.2 (50.5) 14.7% ^b^		49.5 (56.7) 16.0% ^b^	
Jamar Plus	329.2 (95.0)	209.9–494.6	45.2 (50.5) 14.7% ^b^			4.3 (37.4) 1.3%	
Baseline Hydraulic	333.5 (103.6)	176.5–575.7	49.5 (56.7) 16.0% ^b^	4.3 (37.4) 1.3%			
Citec (subgroup, n = 8)	242.0 (29.3)	202.7–296.0		11.7 (27.2) 4.7%		6.5 (29.4) 2.7%	
Jamar Plus (subgroup, n = 8)	253.7 (37.8)	209.9–310.9	11.7 (27.2) 4.7%			5.2 (30.5) 2.1%	
Baseline Hydraulic (subgroup, n = 8)	248.5 (28.2)	215.8–307.3	6.5 (29.4) 2.7%	5.2 (30.5) 2.1%			

^a^ Significant difference with the Citec hand-held dynamometer (*p* = 0.049) and Nicholas hand-held dynamometer (*p* = 0.021); ^b^ Significant difference with the Baseline Hydraulic and Jamar Plus hand-held dynamometers (*p* < 0.01); SD, standard deviation.

## Data Availability

For data supporting the reported results, please contact the corresponding author.

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
