# Peer review of "Variability between Different Hand-Held Dynamometers for Measuring Muscle Strength"

_sensors, 2024, doi:10.3390/s24061861_

Round 1

Reviewer 1 Report

Comments and Suggestions for Authors

Thank you for your work and for the opportunity to review this paper.

I found the article interesting and the methodology appropriate, but I think that if these modifications are valued, the quality of the publication could improve.

  1. Recruitment of Healthy Subjects: It would indeed be beneficial for the authors to provide a detailed description of how they recruited the healthy subjects and their geographical origin. Transparency in recruitment methods enhances the study’s validity.
  2. Abbreviations in Tables: Including explanations for abbreviations (such as SD for standard deviation) in the table footnotes would improve clarity for readers. It’s essential to ensure that readers can interpret the data accurately.
  3. Sample Size Justification: You’ve rightly pointed out that the small sample size (30 subjects) warrants further justification. The authors should explain why this specific sample size was chosen and consider discussing the statistical power of their study.
  4. Additional Citations: The article could benefit from referencing relevant studies, such as the one by Bohannon RW on measuring muscle strength. Including a broader range of references strengthens the literature review.

Reviewer 2 Report

Comments and Suggestions for Authors

Generally, this article investigated the variability among different hand-held dynamometers used to measure muscle strength in healthy individuals, with focus on four muscle groups (upper and lower limbs). In the study, six commercially available dynamometers were compared and grip strength measurements were found to differ between them. There was emphasis placed on the use of interchangeable dynamometers to maximize the use of normative reference values in assessing upper and lower limb strength accurately, which assisted in diagnosing pathology and evaluating treatment outcomes. Accordingly, the study concluded that reliable tools are required to assess muscle strength in patients with neuromuscular disorders and that normative reference values are essential to clinical assessment.

Below are a few minor comments:

On Line 185, it is clearly stated that "To assess hip external rotation ... ", as well as the assessment of the figure shown, however inconsistent descriptions and the inaccurate term "internal" can be found on Line 21, 193, 245, 249, 251, Table 269 as well.

Reviewer 3 Report

Comments and Suggestions for Authors

The article is well written and structured. A few comments.

I suggest the authors bring forward to the paragraph “Study design and participants” the information on the participants, which is now in the paragraph “Variability”.

Thirty participants is a good number; however, it could be interesting to evaluate sample size for a given confidence interval and maximal error using the data from the pilot study conducted on 10 participants. This data will also help for evaluating the subgroup analysis (n=8).

It is evident from Table 1 that the unit range is different among the six hand-held dynamometers used in the study. Is data on accuracy and minimum detectable variation also available? I think it would be interesting for Sensors readers to have more technical details about the different dynamometers.

Table 2 is not always readily readable, in particular for the percentage differences between dynamometers. How should the reader interpret the data in the rows and in the columns? From what I understand the row should allow for reading differences with respect to that particular dynamometer, so for example in the first row for Citec the percentage difference with Nicholas is (98.1-95.5)/98.1=2.7%, that with Commander (98.1-93.2)/98.1=5.0% and so on. However, in the row for Commander the error with Citec should be estimated as (93.2-98.1)/93.2=5.3%. Did the authors average the two values or did they always take the larger value (98.1 N in this case) to calculate the percentage error?

Also, it is not entirely clear to me why the significant difference, marked with superscript a and b, is shown only for the data in the rows even though they are in fact reported similarly.

 In line 149, the Authors specify that the “make” technique was used in the study to measure muscle strength. In line 289, they write that [27] used the “break” test. I believe some explanations on these terms could help readers who are unfamiliar with these tests.
